# Shape Optimization of the Streamlined Train Head for Reducing Aerodynamic Resistance and Noise

**Mengge Yu [1], Jiali Liu [2,*], Wei Huo [1] and Jiye Zhang [3]**

1   College of Mechanical and Electrical Engineering, Qingdao University, Qingdao 266071, China
2   CRRC Qingdao Sifang Co., Ltd., Qingdao 266111, China
3   State Key Laboratory of Traction Power, Southwest Jiaotong University, Chengdu 610031, China
*   Correspondence: liujiali0612@163.com; Tel.: +86-532-6895-3590

**Abstract:** Aiming to improve the comprehensive aerodynamic performance of a high-speed train, a multi-objective shape optimization method for a streamlined train head is proposed in this work. The shape of the streamlined train head is parameterized with some spline curves. The optimization design variables are uniformly sampled using the optimal Latin hypercube design method. The aerodynamic resistance and dipole noise sources are chosen as the optimization objectives, which can be obtained through the computational fluid dynamics (CFD) method. An approximate calculation model is established by the radial basis function neural network so as to effectively predict the values of optimization objectives. The error between the predicted values and actual values of the aerodynamic resistance is less than 1%, and that of the dipole noise source is less than 3 dB, which demonstrate the validity of the approximate calculation model. In the optimization process, the algorithm NSGA-II is adopted to update the values of the optimization design variables, and the approximate calculation model is used to calculate the optimization objectives, which greatly reduces the optimization computation time of the streamlined head shape. Through iterative computation of the optimization algorithm in the design space, each optimized design variable shows a trend of convergence, and the aerodynamic resistance and dipole noise source generally show a decreasing trend. The Pareto front is corrected by the CFD method after optimization. The aerodynamic resistance can be reduced by up to 4.5%, and the dipole noise source can be reduced by up to 3.9 dB.

**Keywords:** high-speed train; optimization; aerodynamic resistance; dipole noise source

## 1. Introduction

High-speed trains have attracted worldwide attention because of their advantages, such as rapid speed, riding comfort, running safety and reliability. However, the operation of high-speed trains also brings considerable problems to the environment, railway construction and locomotive manufacturing industry. The dynamic environment of regular trains is primarily mechanical action, while that of high-speed trains is primarily aerodynamic action. The interaction between trains and the air flow will become more significant with the increase of the cruising speed, resulting in a series of aerodynamic problems, such as aerodynamic resistance, aerodynamic noise, passing pressure wave, tunnel compression wave and expansion wave, crosswind stability [1–4], etc. The aerodynamic resistance of a train is found to be proportional to the square of velocity, and the aerodynamic noise proportional to the sixth power of velocity. This is an objective law that cannot be avoided by any form of ground transportation. If the velocity reaches 300 km/h, the aerodynamic resistance accounts for more than 75% of the total resistance of a high-speed train [5]. Meanwhile, aerodynamic noise has also become the main noise source [6]. Aerodynamic resistance and aerodynamic noise, which are closely related to energy consumption and environmental protection, have become the main factors affecting the improvement of train speed.

In the past decades, numerous investigations have been conducted to reduce the aerodynamic resistance or reduce the aerodynamic noise. Geometric modifications of train shape are the most effective methods to optimize aerodynamic performance. Liu designed some bottom deflectors around the bogie cabin for the purpose of reducing the aerodynamic resistance [7]. Li set up a train model with bionic round pits to investigate the effects of non-smooth surface on the aerodynamic resistance reduction [8]. Hwang proposed two side skirts so as to maximize resistance reduction, and the wind tunnel tests demonstrate that the drag coefficient was reduced by at least 5% [9]. As to the reduction of aerodynamic noise, Zhang established an effective method for calculating aerodynamic noise of a train and designed local structures to reduce the aerodynamic noise [10]. Li investigated the aerodynamic noise of the pantograph of a train and found that the main noise source of pantograph was distributed in the pan-head [11]. Kim investigated the influence of pantograph cavity on the aerodynamic noise of a train as well as the flow around the train, and found that the pantograph cavity configuration could reduce the noise of pantograph [12]. Although these studies provide effective methods and suggestions for the resistance reduction or noise reduction of high-speed trains, the designs in these studies are essentially optimum seeking methods. In this method, some conceptual configurations are first drawn up, then compared and selected through experiments and/or simulations, and finally improved according to the operating conditions. This optimum seeking method relies too much on engineering experience, and the final design is usually not optimal.

To resolve this, a direct optimization method is developed. The optimization design variables are obtained through parametric modeling of train shape, which can be automatically updated by the optimization algorithm, so as to obtain the optimal train shape. Muñoz-Paniagua established a parametric model of the train nose and obtained the optimal train nose shape with the lowest aerodynamic resistance through genetic algorithms [13]. Zhang optimized the train shape based on the Kriging model with the goal of reducing the resistance and lift [14]. Wang considered aerodynamic resistance and lift as optimization objectives and used partial differential equation (PDE) parametric modeling to optimize the train shape [15]. These multi-objective or single-objective direct optimizations for the train head mentioned above are mainly aimed at drag reduction or lift reduction, without considering noise reduction simultaneously. From the perspective of energy saving and environmental protection, we take the aerodynamic resistance and dipole noise source of head car as optimization objectives, and propose an efficient multi-objective optimization method for streamlined train head. The three-dimensional parametric model of the train head is first established. The train aerodynamic model is then described and verified. The optimal Latin hypercube sampling (OLHS) is adopted for uniform sampling in the design space of optimized design variables. An approximate calculation model of train aerodynamics is then set up based on radial basis function (RBF) neural networks. At last, the shape optimization of the streamlined train head is conducted with the NSGA-II algorithm.

## 2. Multi-Objective Optimization Process

### 2.1. Method

The multi-objective optimization problem usually consists of $n$ design variables, $m$ objective variables, and $k$ constraint equations. Its mathematical expression can be expressed as

$$\begin{aligned} \min \quad & \mathbf{y} = (f_1(\mathbf{x}), f_2(\mathbf{x}), \ldots, f_m(\mathbf{x})) \\ s.t. \quad & g_i(\mathbf{x}) \leq 0, i = 1, 2, \ldots, k \end{aligned} \tag{1}$$

where

$$\begin{aligned} & \mathbf{x} = (x_1, x_2, \ldots, x_n) \in \mathbf{X} \\ & \mathbf{y} = (y_1, x_2, \ldots, y_m) \in \mathbf{Y} \\ & \mathbf{X} = \{(x_1, x_2, \ldots, x_n) | \ell_i \leq x_i \leq u_i, i = 1, 2, \ldots, n\} \\ & \mathbf{L} = (\ell_1, \ell_2, \ldots, \ell_n) \\ & \mathbf{U} = (u_1, u_2, \ldots, u_n) \end{aligned} \tag{2}$$

where $\mathbf{X}$ are design variables; $\mathbf{L}$ are lower bounds; $\mathbf{U}$ are upper bounds; $\mathbf{Y}$ are objective variables.

Multi-objective optimization often has mutually concurrent objectives. The promotion of one optimization objective might lead to deterioration of other optimization objectives. To solve multi-objective optimization problem is to make trade-offs and compromises among the optimization objectives, so that each optimization objective can be optimized to the greatest extent. To achieve this, V. Pareto put forward the concept of Pareto's optimal solution set. Suppose $\mathbf{x} \in \mathbf{X}$, if another $\mathbf{x}' \in \mathbf{X}$ does not exist, which makes $f_m(\mathbf{x}') \leq f_m(\mathbf{x})$, $m = 1, 2, \ldots, M$, and at least one of these strict inequalities holds, $\mathbf{x}$ is then called a Pareto optimal solution, the set of which is called the Pareto-optimal solution set. The image of the Pareto optimal solution set in the objective function space is called the Pareto optimal front.

### 2.2. Calculation Procedure

The automatic multi-objective optimization design process of a train head is shown in Figure 1a. In each iteration step, the three-dimensional parametric geometric model is updated on the basis of design variables. The computational mesh is then generated, and the train aerodynamic calculation is conducted so as to acquire the objective variables. After that, the convergence of objective variables is judged. If convergence is achieved, the multi-objective optimization calculation is stopped; and if not, the multi-objective optimization algorithm will be used to update the design variables and conduct calculation of the next iteration step.

As can be seen from the optimization process, the train aerodynamic calculation is required in each optimization iteration step, which is very time-consuming. Aiming at reducing design time of shape optimization, less computational effort of train aerodynamics is required. An approximate calculation model of train aerodynamics, which meets the requirements of engineering accuracy, needs to be developed. Then in each iteration, the approximate calculation model can be used to obtain the value of objective variables, reducing the meshing time and aerodynamic calculation time, which can greatly reduce the total calculation time of optimization design. Therefore, before optimization, an approximate model for computing the train aerodynamics is constructed on the basis of OLHS and RBF neural network in this study. Specifically, the OLHS is adopted to uniformly sample in the optimization design variable space, and the parametric model is carried out according to sampling points to obtain the train model file. According to the model file, the computational grid is generated, and the grid file is obtained. Using the grid file, the train aerodynamic calculation is carried out, and objective variables are obtained. An approximate calculation model between the design variables and objective variables is established by the RBF neural network. The design process of streamlined head shape based on the aerodynamic approximate calculation model is shown in Figure 1b.

### 2.3. Optimal Latin Hypercube Sampling

To obtain a more accurate approximate calculation model, a set of uniformly distributed input variables is required so that the approximate calculation model can better estimate the response of any point. The random Latin hypercube design that emerged in the late 1970s allows for a more effective "space filling" through which more uniform design space points can be obtained. Compared with the traditional experimental design method, the random Latin hypercube design obtains test points that can better fill the entire design space with a smaller number of test points. Nevertheless, the less uniform distribution of design points still exists. Moreover, as the number of levels increases, it is prone to losing some regions in design space. The OLHS improves the uniformity of random Latin hypercube design, making the fitting of input variables and output variables more accurate and realistic [16]. The OLHS enables all test points to be uniformly distributed, with great space filling and equilibrium.

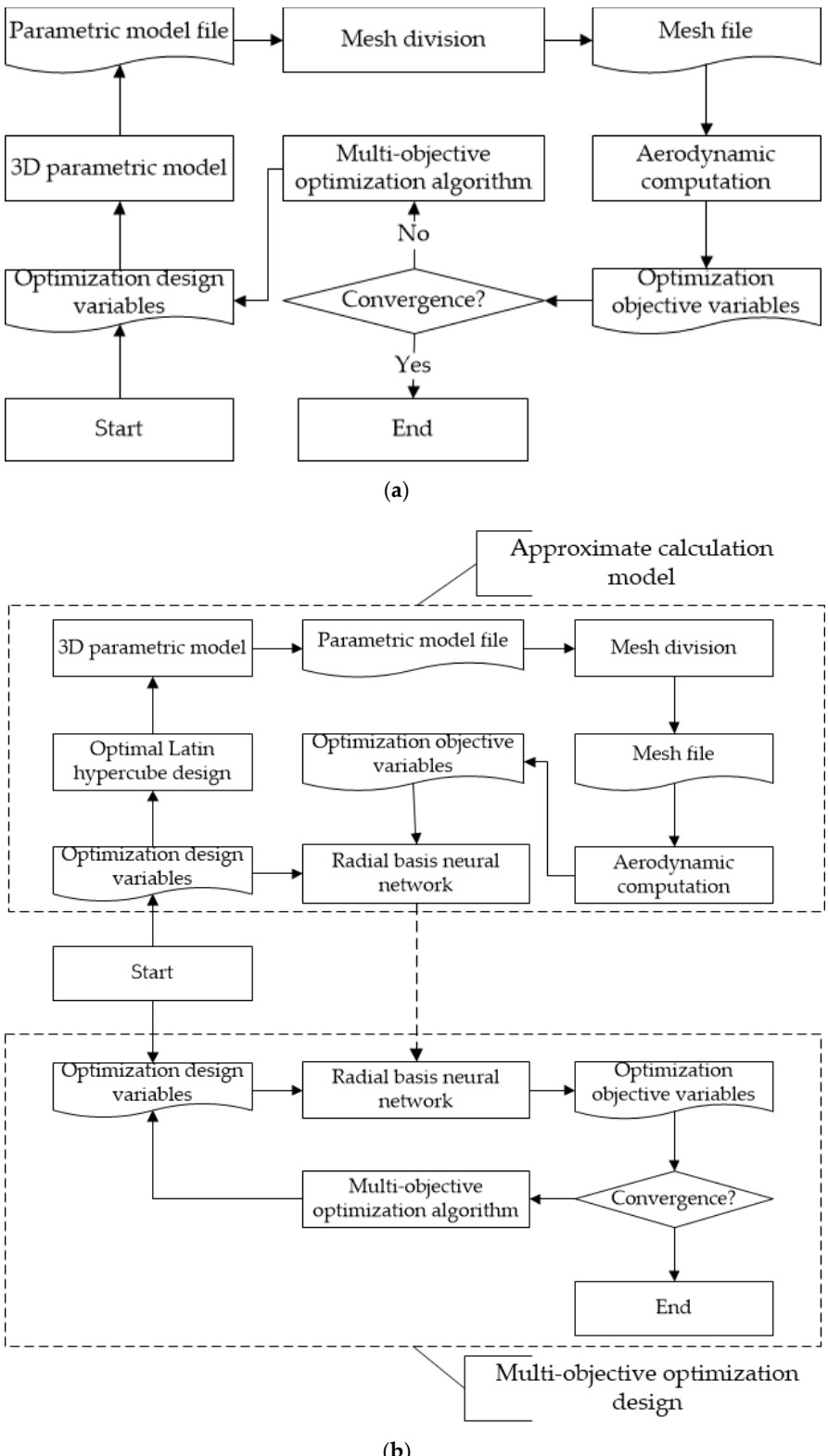

**Figure 1.** Optimization process of streamlined head shape: (**a**) Direct optimization process, (**b**) Optimization process based on approximate model.

The experimental design with N test points and M factors is noted as a $n \times m$ matrix $X = [X_1, X_2, \ldots, X_n]$, each row $X_i^T = [x_{i1}, x_{i2}, \ldots, x_{im}]$ represents an experimental analysis, and each column represents a factor, then the calculation process of OLHS is as follows:

Step 1: An initial design matrix is generated using random Latin hypercube design. In the n-dimensional space, each coordinate interval $\left[x_k^{\min}, x_k^{\max}\right] (k \in [1,n])$ is uniformly divided into m intervals, and the i-the sub-interval is noted as $\left[x_k^{i-1}, x_k^i\right] (i \in [1,m])$. M points are randomly selected to ensure that each level of each factor is studied only once, which constitutes a random Latin hypercube design with M-dimensional space and N samples, denoted as $n \times m$ LHD.

Step 2: Through the update operation of element exchange, a new design matrix is generated.

Step 3: Calculate the optimal conditions for space filling. There are three optimal criteria for space filling that can be selected. In this study, the mini-max distance criterion $\min\limits_{1 \le i,j \le n, i \ne j} d(x_i, x_j)$ is adopted. $d(x_i, x_j)$ denotes the distance between sampling points $x_i$ and $x_j$, which can be calculated by

$$d(x_i, x_j) = d_{ij} = \sqrt{\sum_{k=1}^{m} \left|x_{ik} - x_{jk}\right|^2} \tag{3}$$

Step 4: If optimal conditions are not satisfied, the improved stochastic evolution algorithm is used to seek global optimal solution.

### 2.4. Radial Basis Function Neural Network

After obtaining the experimental design points of the input variables and corresponding output variables using OLHS, the approximate calculation model between input variables and output variables can be established for the optimization design. The neural network is a mathematical model that simulates the essential property of human brains or natural neural networks. It is composed of a large number of nodes (also known as neurons) that are connected to each other. The neural network achieves the purpose of information processing through the interconnection between a large number of nodes. Neural network models have good learning ability, associative storage ability, and high-speed optimization seeking ability and they have been widely used in complex function approximation, pattern recognition, dynamical systems, artificial intelligence, and function optimization. The RBF neural network model has excellent properties in complex function approximation and requires relatively fewer neurons to obtain good approximation results [16].

RBF neural network is a three-layer forward network including input layer, intermediate layer and output layer. The input signals are received by the input layer, and the signals are output by the output layer. The intermediate layer has no direct relationship with the input and output. The RBF neural network takes the Euclid norm of the points to be measured and the sample points as the independent variables, and the radial function as basis function. The neural network model is then established through the linear superposition method.

Suppose that the input layer contains N units, and the input signal $\mathbf{x}$ enters the RBF neural network through the input layer; the intermediate layer contains $p$ cells, and input of the $p$-th unit is represented as $h_p = \|\mathbf{x} - \mathbf{c}_p\|$; the output layer contains one unit. Then, the input of the RBF neural network model can be calculated by:

$$g(\mathbf{x}) = \sum_{p=1}^{p} \lambda_p \varphi_p(\mathbf{x}) + \theta = \sum_{p=1}^{p} \lambda_p \varphi\left(\|\mathbf{x} - \mathbf{c}_p\|\right) + \theta \tag{4}$$

where $\left\{\mathbf{c}_p\right\}_{p=1}^{P} \subset R^N$ denotes the center of the basis function, $c_p$ denotes the center of the $p$-th basis function, $\lambda$ denotes the weight coefficient, $\varphi$ denotes the nonlinear basis function, and $\theta$ denotes the threshold.

The input layer to the intermediate layer is a fixed nonlinear transformation, through which the input signal is mapped to a new space. This mapping relationship depends on the central point of RBF. The intermediate layer to the output layer is a linear transformation.

The output layer is combined in a new linear space with linear weights, where the weights are adjustable parameters of the network.

The commonly used nonlinear basis function is the Gaussian basis function, which has the following expression:

$$\varphi_p(\mathbf{x}) = \exp\left(\frac{\|\mathbf{x} - \mathbf{c}_p\|^2}{2\sigma_p^2}\right) \tag{5}$$

where the parameter $\sigma_p$ denotes the "width" or "flatness" of the $p$-th Gaussian basis function $R_p(\mathbf{x})$, and its value is

$$\sigma_p = \frac{1}{M_p}\sum\|\mathbf{x} - \mathbf{c}_p\|^2 \tag{6}$$

where $M_p$ denotes the number of samples in the $p$-th unit.

### 2.5. Optimization Algorithm

Normalized methods and non-normalized methods are the two main types of solutions to multi-objective optimization problems. Normalized methods solve the multi-objective problems by transforming into single objective problems, which are sensitive to the shape of Pareto optimal front and have a low efficiency in solution. The non-normalized method is an optimization technology that directly deals with multiple objectives using the Pareto mechanism. Hence, the drawbacks of the normalized method are solved. The non-normalized method can make the front of the solution set close to the Pareto front. The representative method of the non-normalized method is the multi-objective genetic algorithm.

The algorithm NSGA-II is adopted to perform the aerodynamic optimization of the streamlined train head. NSGA-II employs non-dominated sorting with elite strategy, and uses a simple crowding operator to maintain population diversity [17]. In the evolutionary process, the population P is first genetically manipulated to obtain the population Q. After merging the two populations, non-dominated sorting and crowding distance sorting are carried out to form a new population P. This process is repeated until the end.

### 3. Train Aerodynamic Model

### 3.1. Parametric Model of Streamlined Head

Because of the good symmetry of the streamlined head shape, only half of the streamlined head needs to be parameterized. The streamlined head is very complex, which is composed of several sub-surface pieces through certain continuous order splicing. In this paper, the left half of the streamlined head shape is composed of B-spline surfaces. Control points are firstly established on the streamlined head, 12 B-spline curves are established from control points, and 7 B-spline surfaces are finally established from these 12 B-spline curves. Thus, the left half of the streamlined head shape is established [18], which is shown in Figure 2. On the basis of Figure 2, the three-car formation train model will be easily established.

In Figure 2, five design variables are extracted, which correspond to the longitudinal line C1, the horizontal contour lines C2 and C3, the central control line C4, the control line of nose height C5.

Lines C1, C2, and C3 have similar deformation form. The deformation of C1 is achieved by modifying values of vertical coordinates of control points, while the deformations of C2 and C3 are achieved by modifying values of horizontal coordinates of control points. Specifically, increase vertical coordinate of the longitudinal midpoint of C1 by $x_1$ ($x_1 > 0$ indicating that this point moves upward, and $x_1 < 0$ indicating that this point moves downward), and vertical coordinates of both ends of C1 do not change. For points between midpoint and endpoints, their vertical coordinates increase according to a linear law. Similarly, the horizontal coordinates of the longitudinal midpoints of C2 and C3 increase by $x_2$ and $x_3$, respectively, where, $x_2 > 0$ or $x_3 > 0$ indicates that the point moves inward. $x_2 < 0$ or $x_3 < 0$ indicates that the point moves outward.

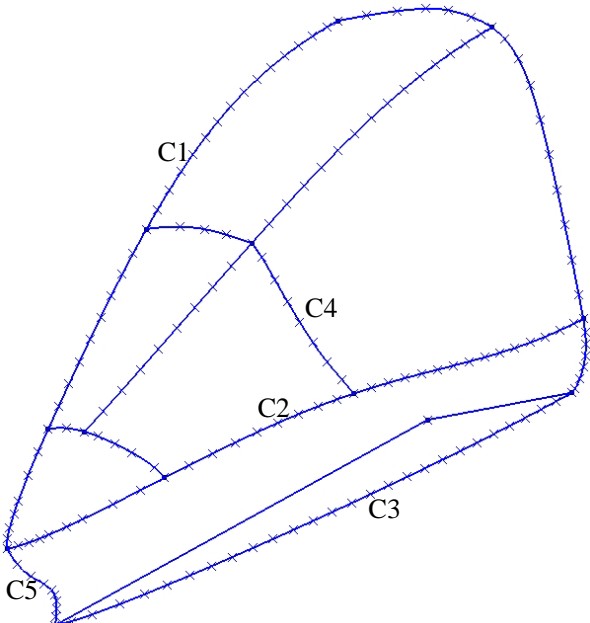

**Figure 2.** Left half of streamlined head.

The deformation of C4 is mainly a deformation of curve concavity and convexity. It is necessary to make the two endpoints of the control line fixed. Meanwhile, the maximum change happens at the midpoint of the control line. The following equation is used here for deformation:

$$y_{4,\text{new}}(i) = y_{4,\text{old}}(i) \times \left(1 + \frac{x_4 \times (i-1) \times (n_4 - i)}{(i-1) \times (i-1) + (n_4 - i) \times (n_4 - i)}\right) \tag{7}$$

where $n_4$ denotes the number of control points on C4, $y_{4,\text{old}}(i)$ denotes the value before deformation, $y_{4,\text{new}}(i)$ denotes the value after deformation, and $i$ denotes $i$-th control point.

For the change of nose height, it is only necessary to multiply the vertical coordinate of the spline curve C5 by a factor $x_5$, when $x_5 > 1.0$, nose height becomes larger; when $x_5 < 1.0$, nose height becomes smaller.

It should be noted that curves associated with deformed curves should also be deformed accordingly to ensure continuity and smoothness of the surface.

When each optimization variable is specified, the coordinates of control points are modified by the self-programming process to obtain a new streamlined head shape. Table 1 shows the value range of each optimization variable and the corresponding deformation. The second row in Table 1 shows the streamlined head shapes for each optimization variable at the lower bound, and the third row shows the ones at the upper bound.

### 3.2. Computational Model

The three-dimensional parametric model described in Section 3.1 is used to set up a train model with three cars. The computational domain as well as the boundary conditions is shown in Figure 3. The front of the computational domain is set as velocity inlet boundary, and the rear is set as pressure outlet boundary. The two sides and top of the domain are set as symmetry boundaries. The bottom of the domain is set as sliding wall boundary, and the train is set as the stationary wall. The train speed in the present paper is 300 km/h. Thus, the flow around the train is considered as in-compressible steady flow. The commercial software Fluent is used to simulate the flow field of the train, the standard $k$-$\varepsilon$ turbulence model is used to resolve the turbulent flow, and the standard wall function is used to resolve the boundary layer.

**Table 1.** Range of design variables and corresponding deformation.

| $x_1$, [−400, 400]/(mm) | $x_2$, [−100, 200]/(mm) | $x_3$, [−200, 200]/(mm) | $x_4$, [−0.2, 0.4] | $x_5$, [−0.8, 1.2] |
| --- | --- | --- | --- | --- |

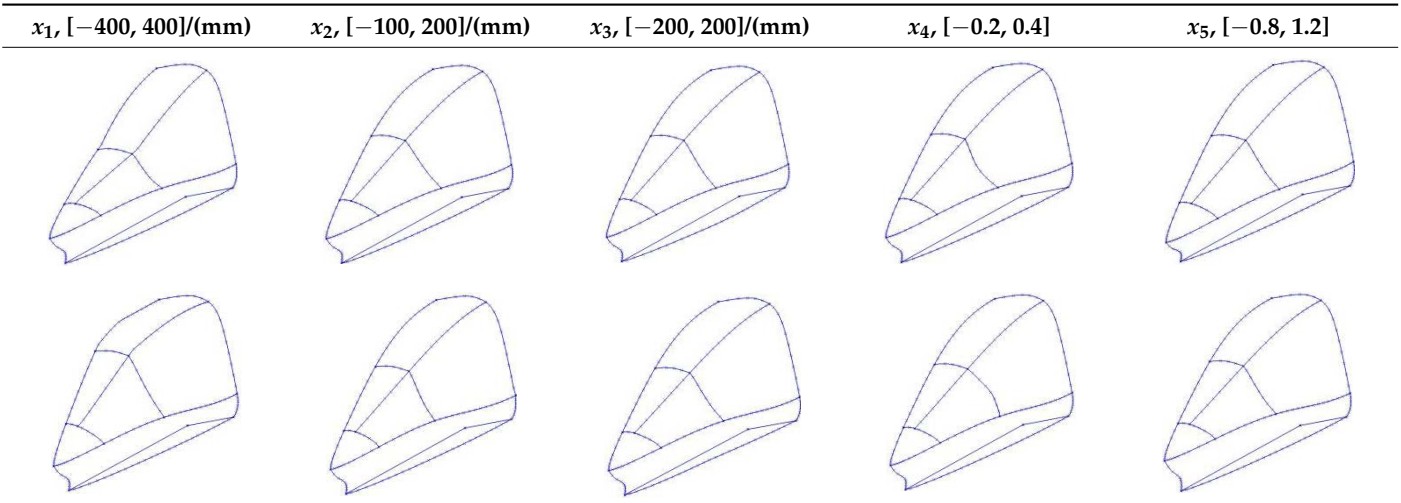

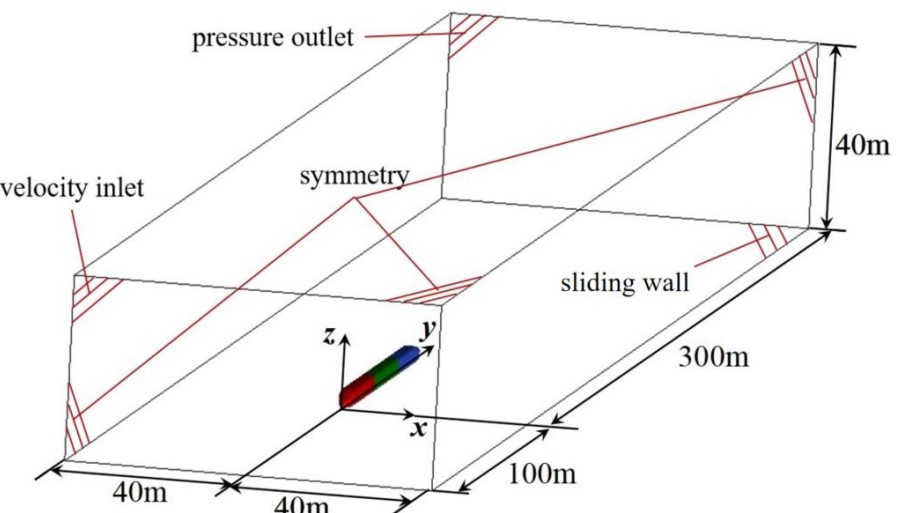

**Figure 3.** Computational domain and boundary condition.

The aerodynamic resistance of a train is directly obtained by aerodynamic calculation, and the aerodynamic noise can be described by dipole noise source on the train surface. Curle [19] analyzed the noise generated by gas flowing through solid surface, which can be expressed as

$$p'(\mathbf{z}, t) = \frac{1}{4\pi a_0} \int_S \frac{(z_i - y_i)n_i}{r^2} \frac{\partial p(\mathbf{y}, \tau)}{\partial t} dS(\mathbf{y}) \tag{8}$$

where $\mathbf{y}$ and $\mathbf{z}$ are the positions of near-field points and far-field points, respectively; $y_i$ and $z_i$ are components of $\mathbf{y}$ and $\mathbf{z}$; $t$ is time; $\tau$ is delay time, $\tau = t - r/a_0$; $p'$ is sound pressure; $a_0$ is sound velocity; $S$ is solid surface; $r$ is the distance between $\mathbf{y}$ and $\mathbf{z}$; $n_i$ is the unit normal vector component of $S$.

Based on Formula (8), the total acoustic power $P_A$ emitted from the entire body surface can be computed from

$$P_A = \int_S \frac{A_c(\mathbf{y})}{12\rho_0 \pi a_0^3} \left[ \overline{\frac{\partial p(\mathbf{y}, \tau)}{\partial t}} \right]^2 dS(\mathbf{y}) \equiv \int_S I(\mathbf{y}) dS(\mathbf{y}) \tag{9}$$

where $A_c$ is the correlation region and $I(\mathbf{y})$ is the dipole noise source on the solid surface, which can be interpreted as the contribution of the noise per unit area of solid surface to the total sound pressure.

The sound power level $L_p$ is defined as

$$L_p = 10 \log(P_A / P_r) \tag{10}$$

where $P_r$ is reference sound power, $P_r = 10^{-12} \text{W}/\text{m}^3$.

After the flow filed of the train is computed, the broadband noise source module in the commercial software Fluent is activated. Then, the dipole noise source on the streamlined head can be obtained.

Grid-independence test is conducted aiming at eliminating the influence of grid density on flow field simulation. Three sets of mesh with different grid density are generated, and the information of grid setting and calculation results are shown in Table 2. The three meshes have the same settings of boundary layers. The height of first layer is 2 mm, the growth ratio is 1.1, and there are 8 layers in total. As presented in Table 1, when the maximum size of the surface grid is densified from 250 mm (mesh1) to 200 mm (mesh2), the aerodynamic resistance decreases by 4.55%, and the dipole noise source increases by 0.5 dB. When the maximum size of the surface grid is densified from from 200 mm (mesh2) to 150 mm (mesh3), the aerodynamic resistance increases by 0.86%, and the dipole noise source increases by 0.1 dB. Therefore, mesh2 is chosen for the subsequent aerodynamic calculations.

**Table 2.** Grid-independence tests.

| | Maximum Surface Grid | Total Cells (Million) | Aerodynamic Resistance(N) | Dipole Noise Source(dB) |
|---|---|---|---|---|
| mesh1 | 250 | 4.18 | 3456.2 | 110.7 |
| mesh2 | 200 | 5.92 | 3298.9 | 111.2 |
| mesh 3 | 150 | 8.56 | 3327.3 | 111.3 |

## 4. Results

The OLHS was used to obtain experimental design points of each input variable, which is shown in Figure 4. As plotted in Figure 4, each design variable has a relatively uniform distribution in its design space. Figure 5 shows the objective variables corresponding to the experimental design points, which are the aerodynamic resistance and the dipole noise source of a high-speed train.

The RBF neural network was used to establish the approximate calculation model between input variables and output variables. Figure 6 presents the differences between the predicted and actual values of the aerodynamic resistance and dipole noise source, where Figure 6a indicates the percentage error of aerodynamic resistance, and Figure 6b indicates that of the dipole noise source. As shown in Figure 6, the percentage error between the predicted values and actual ones of aerodynamic resistance is less than 1%, and that of the dipole noise source is less than 3 dB. Thus, the approximate calculation model established by the RBF neural network has a good approximation effect.

In the optimization calculation, the initial sampling points of the NSGA-II algorithm are set to be 20, and 50 generations of genetic calculation are performed. Thus, the optimization is completed after completion of 1000 designs. Figure 7 depicts histories of design variables during optimization, and the pattern "★" in the figure indicates the Pareto optimal solution. As plotted in Figure 7, each optimization design variable shows a convergence trend through the sampling of the optimization algorithm. The NSGA-II algorithm will find the Pareto optimal design point that minimizes the objective value through iterative computation. The variable $x_5$ does not seem to converge. Nevertheless, this variable is basically irrelevant to the aerodynamic resistance and dipole noise source according to the correlation analysis.

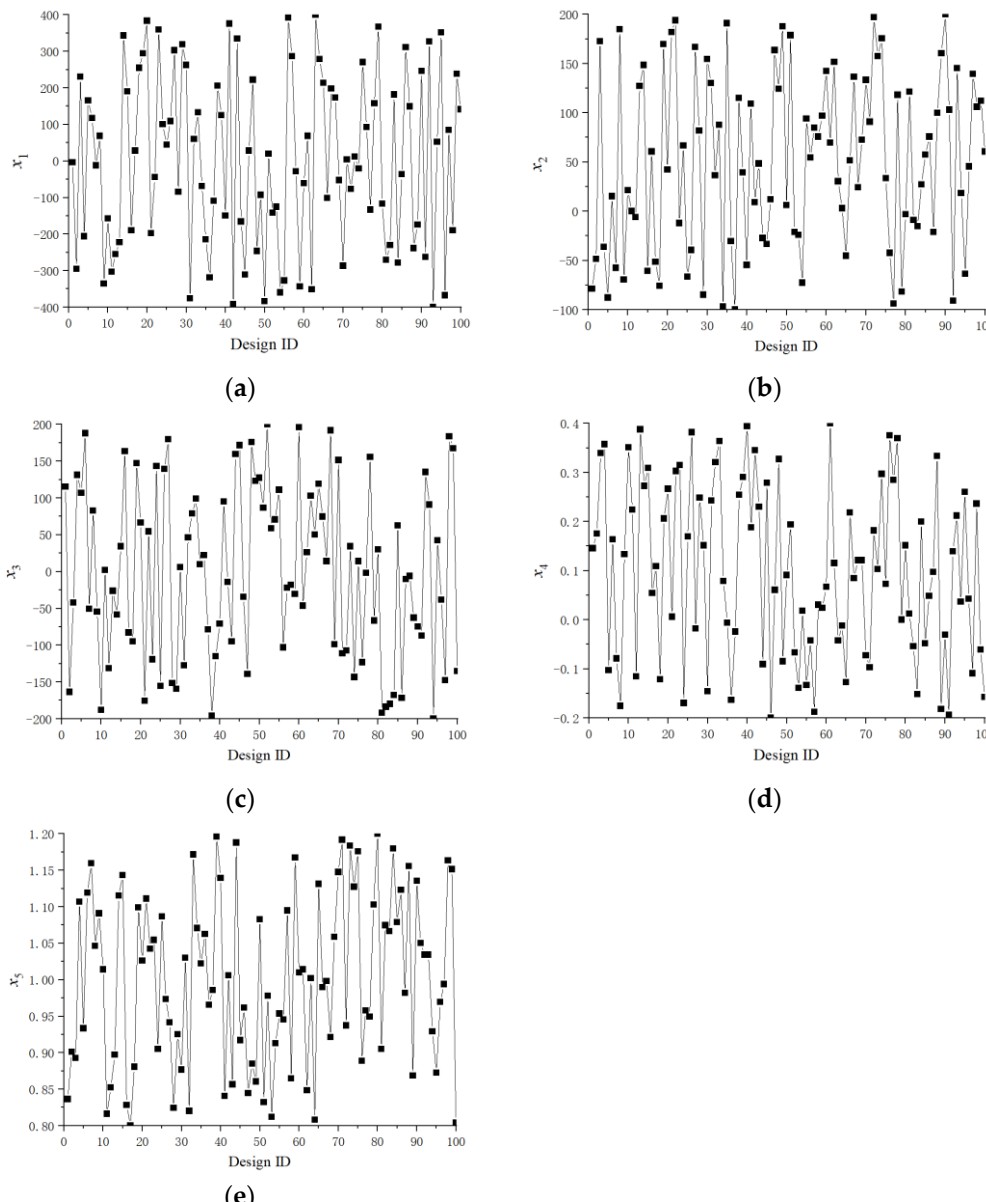

**Figure 4.** Experimental design points for each input variable: (**a**) $x_1$, (**b**) $x_2$, (**c**) $x_3$, (**d**) $x_4$, (**e**) $x_5$.

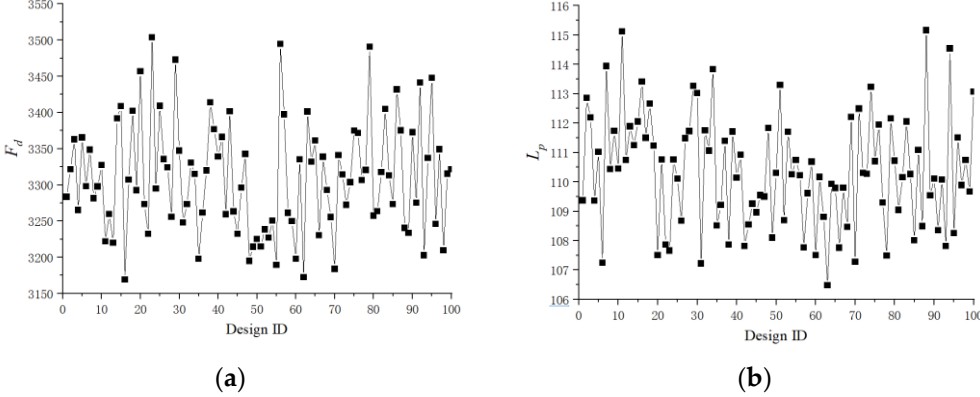

**Figure 5.** Output variables at each design point: (**a**) Aerodynamic resistance, (**b**) Dipole noise source.

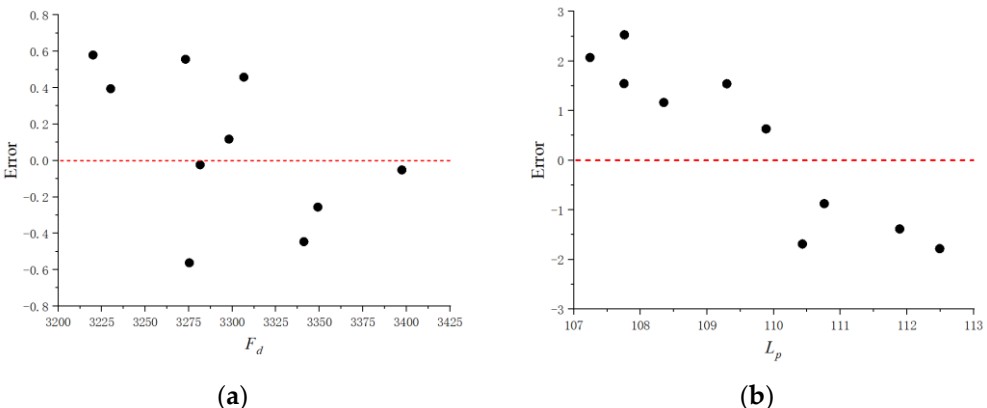

**Figure 6.** Error between predicted value and actual value of output variables: (**a**) Percentage of aerodynamic resistance, (**b**) Difference of dipole noise source.

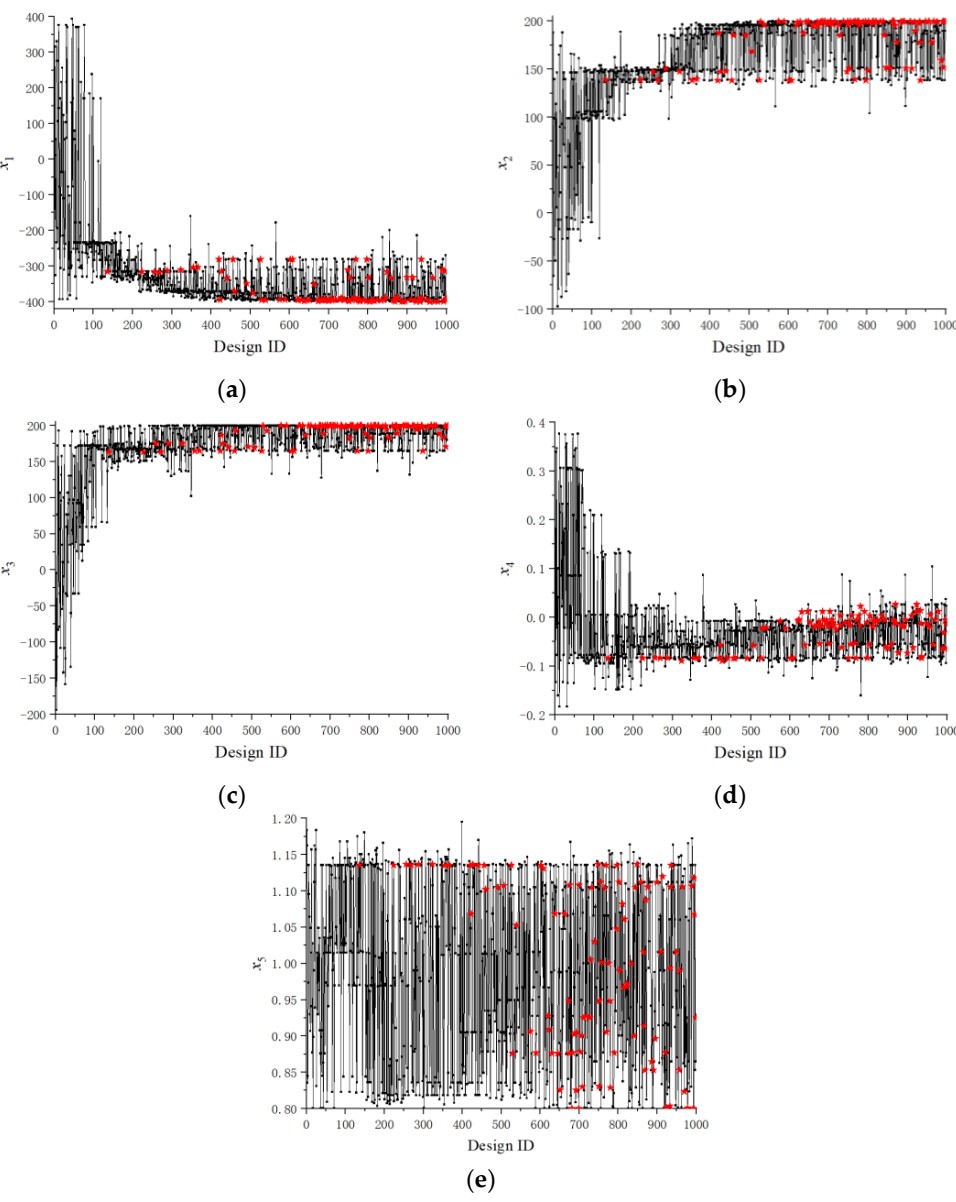

**Figure 7.** Histories of optimization design variables: (**a**) $x_1$, (**b**) $x_2$, (**c**) $x_3$, (**d**) $x_4$, (**e**) $x_5$.

Figure 8 shows histories of objective variables. The pattern "★" indicates the value of objective variable on the Pareto front. As plotted in Figure 8, through the sampling of the optimization algorithm in the design space, the time histories of aerodynamic resistance and dipole noise source show a decreasing trend. The streamlined head shape is gradually improved towards the optimization of aerodynamic resistance and dipole noise source.

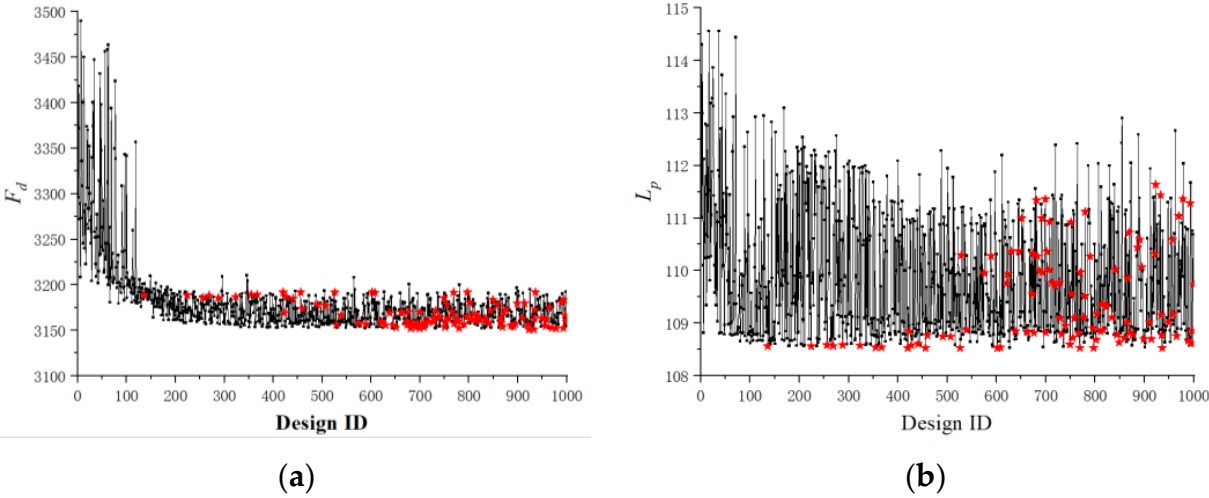

**Figure 8.** Histories of objective variables: (**a**) Aerodynamic resistance, (**b**) Dipole noise source.

The Pareto front is shown in Figure 9. The dot "●" in the figure denotes the aerodynamic resistance and dipole noise source of the initial streamlined head shape. As depicted in Figure 9, both the aerodynamic resistance and dipole noise source have been improved through the optimization. In addition, the Pareto front obtained by the approximate calculation model is not much different from that obtained by the direct optimization, which demonstrates that the approximate calculation model can achieve good optimization results. The optimal aerodynamic resistances for the optimization based on the approximate calculation model and direct optimization are 3150.7 N and 3149.1 N, respectively, which are very close to each other. The optimal dipole noise sources for the optimization based on approximate calculation model and direct optimization are 108.5 dB and 107.2 dB, and the error is 1.3 dB. The aerodynamic resistance is reduced by up to 4.5%, and the dipole noise source is reduced by up to 3.9 dB through optimization, compared with the original streamlined head shape.

Figure 10 presents a comparison between initial head shape and the one with the lowest aerodynamic resistance/dipole aerodynamic noise, in which the black profile line represents the original head type and the red profile line represents the optimized head type. As presented in Figure 10, compared with initial head shape, line C1 of both optimal head shapes is concave downward, and the concave degree of the optimal head shape with the lowest aerodynamic resistance is greater. Lines C2 and C3 move toward the longitudinal symmetry plane, and the variation of the head shape with the lowest aerodynamic resistance is more significant. The main differences between the two optimal head shapes are the changes of C4 and C5. For line C4, the optimal head shape with the lowest aerodynamic resistance does not vary much with respect to the initial shape, while the optimal head shape with the lowest dipole noise source is more concave. For line C5, the nose height of the optimal head shape with the lowest aerodynamic resistance becomes lower, while the one with the lowest dipole noise source becomes higher.

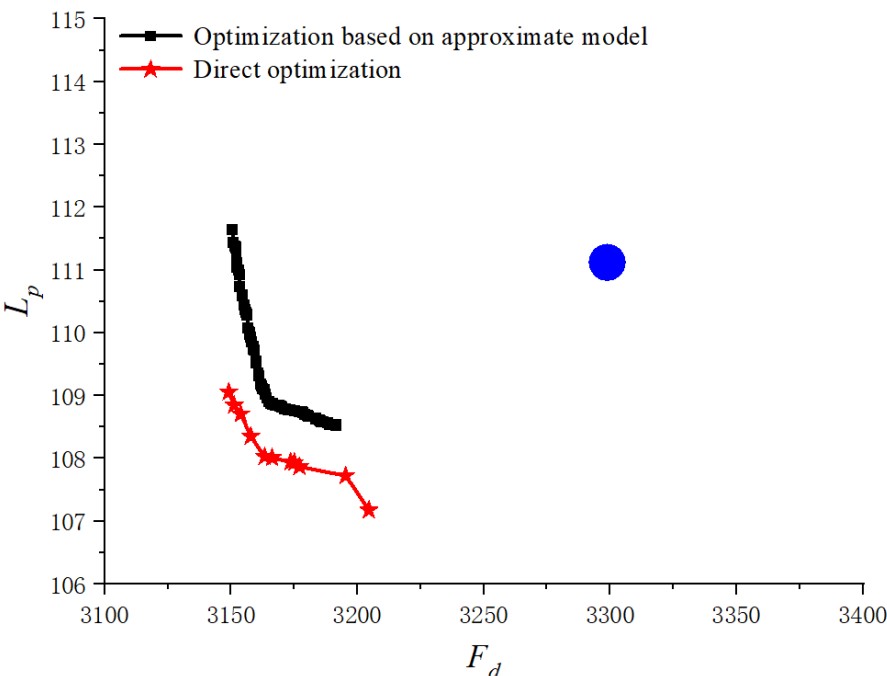

**Figure 9.** Pareto front.

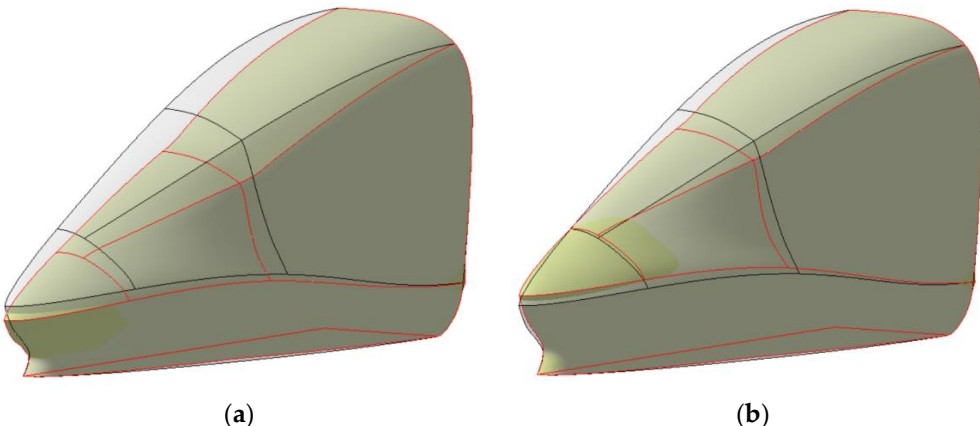

**Figure 10.** Comparison of optimal and original streamlined heads: (**a**) Optimal streamlined head for the aerodynamic resistance, (**b**) Optimal streamlined head for dipole noise source.

## 5. Conclusions

The streamlined head design is one of the core technologies of high-speed trains. In the past, the streamlined head design was mainly based on the optimum seeking method. Thus, it was often impossible to obtain the optimal streamlined head. In this study, the streamlined head of a train model is divided into several B-spline surfaces, and five critical lines are selected to construct the parametric modeling of the streamlined head. For the purpose of energy saving and environmental protection, the aerodynamic resistance and dipole noise source are taken as the objective variables. The aerodynamic approximate calculation model was then established by the OLHS and RBF neural network, which could greatly reduce the calculation amount of train aerodynamics. The multi-objective optimization design of the streamlined head is carried out using the genetic algorithm NSGA-II and several optimal streamlined heads with relatively superior performance of both aerodynamic resistance and aerodynamic noise are obtained. The head shape with the optimal resistance and the one with the optimal aerodynamic noise are finally discussed, and the influence of design variables on the aerodynamic resistance and dipole noise source is further explored. The influence of lines C1, C2, and C3 on aerodynamic

resistance and aerodynamic noise is similar. Nevertheless, the influence of lines C4 and C5 on the aerodynamic resistance and aerodynamic noise is contradictory. The methodology proposed in this study can greatly reduce the design cycle of the streamlined head, allowing to obtain a streamlined head with better aerodynamic performance. After optimization, the aerodynamic resistance is reduced by up to 4.5%, and the dipole aerodynamic noise source is reduced by up to 3.9 dB.

**Author Contributions:** Conceptualization, M.Y.; methodology, M.Y.; software, J.L.; validation, J.L.; formal analysis, W.H.; investigation, J.Z.; resources, J.Z.; writing—original draft preparation, M.Y.; writing—review and editing, M.Y.; visualization, J.L.; supervision, W.H.; funding acquisition, M.Y. All authors have read and agreed to the published version of the manuscript.

**Funding:** This research was funded by the Natural Science Foundation of Shandong Province (Grant No. ZR2022ME180), National Natural Science Foundation of China (Grant No. 51705267), the Postdoctoral Research Foundation of China (Grant No. 2018M630750), the Open Research Foundation of State Key Laboratory of Traction Power (Grant No. TPL2005).

**Institutional Review Board Statement:** Not applicable.

**Informed Consent Statement:** Not applicable.

**Conflicts of Interest:** The authors declare no conflict of interest.

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
