# Peer review of "Shape Optimization of the Streamlined Train Head for Reducing Aerodynamic Resistance and Noise"

_applsci, doi:10.3390/app121910146_

Round 1
Reviewer 1 Report
There is a need for improvement of the paper, with following remarks:
1. Chapters numbering is not valid, with Chapter 5: Conclusions, following chapter 8.
2. In my opinion, the structure of the paper should be as follows:
1. Introduction
2. Multi-objective optimization process
2.1. Method (before Multi-objective optimization method)
2.2. Calculation Procedure (before Multi-objective optimization process; the same as the head title of the chapter)
2.3. Optimal Latin hypercube sampling (before Chapter 5)
2.4. Radial basis function neural network (before Chapter 6)
2.5. Optimization Algorithm (before Chapter 7)
3. Train Aerodynamic model
3.1. Parametric model of streamlined head (before Chapter 3. Parametric model of streamlined head)
3.2. Computational model (before Chapter 4. Train Aerodynamic model)
4. Results
5. Conclusion
I think that is better structure of the paper, so do it like above.
Furthermore:
3. There is no Figure 7, i.e. after Figure 6 there is Figure 8.
4. Put spaces before references [10], [14] and [15], in text.
5. The quotation in the text should be always the same, like "closed" cavity on line 61.
6. After equation (2) put variables: with X are ..., L are ..., and so on,
7. On line 157 there is some error. I suppose there should be x>0 and not x1(x1>0
8. Line 190: as in [17]
9. Line 294: Gaussian ...
10. In the Abstract: The first letter is Bold A
11. Change the the start of the first sentence of the Abstract to something like: "Aiming to improve..."; or "In order to improve ..."; or "With goal to ..."
12. Line 97: Multiobjective optimization often has mutually concurrent objectives.
Overall, change the structure of the paper to be more readable.
Reviewer 2 Report
Line 182 Paragraph 4. The methodology used to derive acoustic radiated noise should be explained better and more in detail. From the text, the following is missing:
What are boundary conditions? What speed?
Where precisely was the noise computed? I suppose in the far-field but this is not specified. Was computed on points on the computational domain?
How exactly was the CFD solution used to derive the acoustic radiated noise?
What solver was used to derive the CFD solution? In-house, commercial? Considering that the authors are using the k-epsilon model, were they using the wall functions to resolve the boundary layer?
Equations 4 and 5 indicate how to compute pressure in the far field but not how to correlate the local pressure distribution to CFD results (for example boundary layer thickness and skin friction see Goody for example).
Line 206. What is the average y+ value for the train surface?
Line 207. Considering standard practice 15 layers seem too few to accurately resolve the boundary layer.
Round 2
Reviewer 2 Report
Reviwer would like to thank the authors for their efforts in improving the paper